# Iterative Causal Discovery in the Possible Presence of Latent Confounders and Selection Bias

**Raanan Y. Rohekar, Shami Nisimov, Yaniv Gurwicz, Gal Novik**
Intel Labs
{raanan.yehezkel, shami.nisimov, yaniv.gurwicz, gal.novik}@intel.com

## Abstract

We present a sound and complete algorithm, called iterative causal discovery (ICD), for recovering causal graphs in the presence of latent confounders and selection bias. ICD relies on the causal Markov and faithfulness assumptions and recovers the equivalence class of the underlying causal graph. It starts with a complete graph, and consists of a single iterative stage that gradually refines this graph by identifying conditional independence (CI) between connected nodes. Independence and causal relations entailed after any iteration are correct, rendering ICD anytime. Essentially, we tie the size of the CI conditioning set to its distance on the graph from the tested nodes, and increase this value in the successive iteration. Thus, each iteration refines a graph that was recovered by previous iterations having smaller conditioning sets—a higher statistical power—which contributes to stability. We demonstrate empirically that ICD requires significantly fewer CI tests and learns more accurate causal graphs compared to FCI, FCI+, and RFCI algorithms (code is available at `https://github.com/IntelLabs/causality-lab`).

## 1 Introduction

Causality plays an important role in many sciences, such as social sciences, epidemiology, and finance (Pearl, 2010; Spirtes, 2010). Understanding the underlying mechanisms is crucial for tasks such as explaining a phenomenon, predicting, and decision making. Pearl (2009) provided a machinery for automating the process of answering interventional and (retrospective) counterfactual queries even when only observed data is available, and determining if a query cannot be answered given the available data type (identifiability). This requires knowledge about the true underlying causal structure; however, in many real-world situations, this structure is unknown. There is a large body of literature on recovering causal relations from observed data—causal discovery (Spirtes et al., 2000; Peters et al., 2017; Cooper & Herskovits, 1992; Chickering, 2002; Shimizu et al., 2006; Hoyer et al., 2009; Rohekar et al., 2018; Yehezkel & Lerner, 2009; Nisimov et al., 2021), differing in the assumptions upon which they rely. In this work we assume a directed acyclic graph (DAG) for the underlying causal structure and focus on learning it from observational data. Furthermore, we assume the causal Markov and faithfulness assumptions, and consider recovering the structure by performing a series of conditional independence (CI) tests (Spirtes et al., 2000). In this setting the true DAG is statistically indistinguishable from many other DAGs. Moreover, when considering the possible presence of latent confounders and selection bias (no causal sufficiency), the true DAG cannot be recovered. Instead, Richardson & Spirtes (2002) proposed the maximal ancestral graph (MAG), which represents independence relations among observed variables, and the partial ancestral graph (PAG), which is a Markov equivalence class of MAGs—a set of MAGs that cannot be ruled out given the observed independence relations.

Recently, causal identification was demonstrated for PAG models (Jaber et al., 2018, 2019), which is a more practical use of these models. That is, by using only observed data and no prior knowledge on the underlying causal relations, some identification and causal queries can be answered.

In this paper, we address the problem of learning a PAG such that interrupting the learning process results in a correct PAG. That is, all the entailed independence and causal relations are correct, although it can be less informative. This anytime property is important in many real-world settings where it is desired to recover as many causal relations as possible under limited compute power.

## 2 Related Work

Causal discovery in the potential presence of latent confounders and selection bias requires placing additional assumptions. In this paper we assume the causal Markov assumption (Pearl, 2009), faithfulness (Spirtes et al., 2000), and a DAG structure for the underlying causal relations. In this setting, several causal discovery algorithms have been proposed, FCI (Spirtes et al., 2000), RFCI (Colombo et al., 2012), FCI+ (Claassen et al., 2013), and GFCI (Ogarrio et al., 2016). Limitations of FCI have been reported previously where it tends to erroneously exclude many edges that are in the true underlying graph, and it requires many CI tests with large conditioning sets. The GFCI algorithm employs a greedy score-based approach to improve the accuracy for small data sizes (small sample). However, it requires additional assumptions for justifying the score function that it uses. The RFCI algorithm alleviates computational complexity by avoiding the last stage of FCI. This stage requires many CI tests having large conditioning sets. Although it is sound (outputs correct causal information), it is not complete (some MAGs in the equivalence class can be ruled out given the data).

Similarily to the FCI and FCI+ algorithms, we consider a procedure that is sound and complete in the large sample limit (or when a perfect conditional independence oracle is used). However, these algorithms, for a MAG, treat nodes that are m-separated by adjacent nodes differently from nodes that are m-separated by a minimal separating set that includes nodes outside the neighborhood. In contrast, we treat all possible separating sets in a similar manner. We employ an iterative procedure that gradually increases the search radius on the graph for identifying minimal separating sets. This allows our method to be interrupted at any iteration, returning a correct graph, similarly to the anytime FCI algorithm (Spirtes, 2001).

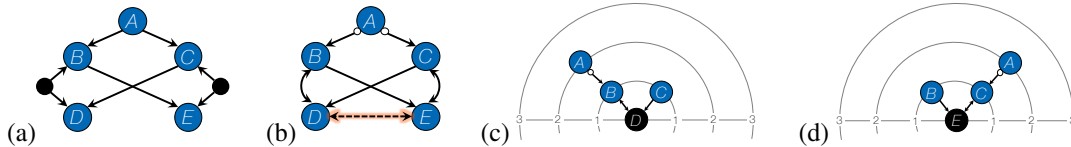

(a)    (b)    (c)    (d)

Figure 1: An example for the construction of conditioning sets by the ICD algorithm. (a) The true underlying DAG, where black circles represent latent variables. (b) A PAG resulting after three ICD iterations, $r \in \{0, 1, 2\}$—a 2-**O-equivalence** class. (c) and (d) PDS-trees with $r = 3$ search radii for nodes $D$ and $E$, respectively, where radial coordinate indicates distance from the root.

## 3 Anytime Iterative Discovery of Causal Relations

First, we provide definitions and assumptions. Then, we describe the iterative causal discovery (ICD) algorithm and prove its correctness. We conclude by discussing efficiency and stability.

### 3.1 Preliminaries

A causal DAG, $\mathcal{D}$, over nodes $\mathbf{V} = \mathbf{O} \cup \mathbf{S} \cup \mathbf{L}$ is denoted by $\mathcal{D}(\mathbf{O}, \mathbf{S}, \mathbf{L})$, where $\mathbf{O}$, $\mathbf{S}$, and $\mathbf{L}$ represent disjoint sets of observed, selection, and latent variables, respectively. We use an ancestral graph (Richardson & Spirtes, 2002) to model the conditional independence relations among the observed variables $\mathbf{O}$ in the causal DAG $\mathcal{D}$. This class of graphical models is useful since for every causal DAG there exits a unique MAG. In this setting, our method is aimed at recovering the MAG of the true underlying DAG, from the result of CI tests. A CI test is commonly a statistical hypothesis test used to determine from observed data whether two variables are statistically dependent conditioned on a set of some other variables (a conditioning set). If independence is found, the conditioning set is called a separating set for the tested nodes. However, in this setting the MAG cannot be fully recovered, and only a Markov equivalence class, represented by a PAG, can be recovered. In a PAG, a variant edge-mark is denoted by an empty circle '—o $X$'. Namely, in the equivalence class there

exists at least one MAG that has a tail edge-mark '——$X$' and at least one MAG that has an arrowhead '—>$X$' at the same location.

**Definition 1** (O-equivalence (Spirtes et al., 2000))**.** *Two DAGs, $\mathcal{D}_i(\mathbf{O}, \mathbf{S}_i, \mathbf{L}_i)$ and $\mathcal{D}_j(\mathbf{O}, \mathbf{S}_j, \mathbf{L}_j)$ are said to be* **O**-*equivalent if and only if*

$$\mathbf{X} \perp\!\!\!\perp \mathbf{Y}|(\mathbf{Z} \cup \mathbf{S}_i) \text{ in } \mathcal{D}_i \iff \mathbf{X} \perp\!\!\!\perp \mathbf{Y}|(\mathbf{Z} \cup \mathbf{S}_j) \text{ in } \mathcal{D}_j,$$

*for every possible disjoint subsets* $\mathbf{X}$, $\mathbf{Y}$, *and* $\mathbf{Z}$ *of* $\mathbf{O}$.

That is, the observed d-separation relations in two **O**-equivalent DAGs, $\mathcal{D}_i$ and $\mathcal{D}_j$, are identical. Spirtes (2001) defines equivalence considering an $n$-**oracle** for testing conditional independence. It returns "dependence" if the conditioning set size is larger than $n$, otherwise d-separation is tested and returned. Thus, $n$-**O**-**equivalence** class is defined by using the $n$-**oracle** instead of d-separation in Definition 1. The PAG that represents this equivalence class is called $n$-**representing** (Spirtes, 2001).

In a MAG, node $X$ is in **D**-**Sep**$(A, B)$ (note the capitalization of 'D') if and only if $X \neq A$ and there is an undirected path between $A$ and $X$ such that every node on the path, except the endpoints, is a collider and is an ancestor of $A$ or $B$ (Spirtes et al., 2000). The FCI algorithm utilizes a super-set of **D**-**Sep**, called **Possible**-**D**-**Sep**$(A, B)$ (Spirtes et al., 2000) for testing CI. It is defined for a skeleton learned by the PC algorithm[1] (Spirtes et al., 2000) and its identified v-structures, referred in this paper as the first stage of FCI. This super-set, used by FCI in its second stage, includes all the nodes connected by a path to $A$, where every node on this path, except the end-points, is a collider or part of a triangle, hiding its orientation. We consider a smaller super-set of **D**-**Sep** for PAGs and take special interest in the path that connects any $Z \in$ **Possible**-**D**-**Sep**$(A, B)$ to $A$.

**Definition 2** (PDS-path)**.** *A possible-D-Sep-path (PDS-path) from $A$ to $Z$, with respect to $B$, in PAG $\mathcal{G}$, denoted $\Pi_B(A, Z)$, is a path $\langle A, \ldots, Z \rangle$ such that $B$ is not on the path and for every sub-path $\langle U, V, W \rangle$ of $\Pi_B(A, Z)$, $V$ is a collider or $\{U, V, W\}$ forms a triangle.*

Note that an edge in a PAG is a PDS-path. Following the definition of PDS-path, we define a tree rooted at a given node consisting of all PDS-paths from that node.

**Definition 3** (PDS-tree)**.** *A possible-D-Sep-tree (PDS-tree) for $A$, with respect to $B$ given PAG $\mathcal{G}$, is a tree rooted at $A$, such that there is a path from $A$ to $Z$, $\langle A, V_1, \ldots, V_k, Z \rangle$, if and only if there is a PDS-path $\langle A, V_1, \ldots, V_k, Z \rangle$, with respect to $B$, in $\mathcal{G}$.*

Lastly, considering the last condition of **D**-**Sep** definition, we define a possible ancestor of a node in a given PAG.

**Definition 4** (Possible Ancestor)**.** *In a PAG, $V_i$ is a possible ancestor of $V_{i+k}$ if there is a path $\langle V_i, V_{i+1}, \ldots, V_{i+k} \rangle$ such that, $\forall j \in \{i, \ldots, i+k-1\}$, on the edge $(V_j, V_{j+1})$ there is no arrowhead at $V_j$ and no tail edge-mark at $V_{j+1}$.*

### 3.2 Iterative Causal Discovery Algorithm

We present a single stage that is called iteratively, for recovering the underlying equivalence class, represented by a PAG, for the true underlying DAG $\mathcal{D}(\mathbf{O}, \mathbf{S}, \mathbf{L})$. Each iteration is parameterized by $r$, where $r \in \{0, \ldots, |\mathbf{O}| - 2\}$. Given a PAG returned by the previous $r - 1$ iteration, each pair of connected nodes $A$ and $B$ is tested for independence conditioned on a set $\mathbf{Z} \subset \mathbf{O}$. If independence is found, the connecting edge is removed. The conditioning set $\mathbf{Z}$ should comply with the following conditions for $(A, B)$ or $(B, A)$, which we call *ICD-Sep conditions*. ICD-Sep conditions for the ordered pair $(A, B)$ are,

1. $|\mathbf{Z}| = r$,
2. $\forall Z \in \mathbf{Z}$, there exists a PDS-path $\Pi_B(A, Z)$ such that,
   (a) $|\Pi_B(A, Z)| \leq r$ and
   (b) every node on $\Pi_B(A, Z)$ is in $\mathbf{Z}$,
3. $\forall Z \in \mathbf{Z}$, node $Z$ is a possible ancestor of $A$ or $B$ (not a necessary condition).

---

[1] PC is a causal discovery algorithm assuming causal sufficiency (absence of hidden confounders and selection bias). Conditioning sets consist of nodes only from the neighborhood of the tested nodes.

Condition 1 restricts the conditioning set size, and condition 2 restricts its distance from the tested nodes. Tying these two conditions together is the key idea of the presented ICD algorithm. Condition 3 follows the last condition in the definition of **D-Sep**. It is not a necessary condition for the correctness of the proposed method but can reduce the number of considered conditioning sets when testing independence. Sets complying with ICD-Sep conditions are denoted **ICD-Sep**.

In comparison to **Possible-D-Sep** that is utilized by the FCI algorithm, **ICD-Sep** is a smaller super-set of $\{\mathbf{S} \mid \mathbf{S} \subset \mathbf{D}\text{-}\mathbf{Sep}, |\mathbf{S}| = r\}$. Nevertheless, **ICD-Sep** and **Possible-D-Sep** may be similar when considering densely connected graphs or when the recoverable equivalence class is large (many variant edge-marks), such as a complete graph. However, these cases are usually rare in real-world scenarios.

Next, consider a PDS-tree for $A$ with respect to $B$ in $\mathcal{G}$. Sets complying with ICD-sep conditions can be created by traversing this tree. Condition 1 restricts the number of nodes to be included in $\mathbf{Z}$ to the specific value of $r$. Condition 2 places constraints on the nodes in $\mathbf{Z}$ as follows. Condition 2a limits the radius around $A$ in which the nodes of $\mathbf{Z}$ reside, effectively limiting the depth of the PDS-tree. Condition 2b ensures that if a node is in $\mathbf{Z}$, then every node on the PDS-path connecting it to $A$ is also in $\mathbf{Z}$. An example is given in Figure 1, where (a) is the true underlying DAG and (b) is the corresponding PAG that represents a 2-**O-equivalence class**. A redundant edge between $D$ and $E$ exists. PDS-trees for $D$ and $E$ are depicted in Figure 1 (c) and (d), respectively. An example for complying with condition 2b when constructing an **ICD-Sep** for $(D, E)$ is as follows. From Figure 1 (c) if $A$ is in the set, then $B$ must also be in the set, as it lies on the only path from $D$ to $A$.

It is important to note that by parameter $r$, we bind the conditioning set size to its distance from the tested nodes (ICD-Sep conditions 1 and 2a). That is, the conditioning set size is bounded by the shortest PDS-path length connecting its nodes to the tested nodes.

In an ICD iteration $r$, first, all connected edges are tested for independence conditioned on **ICD-Sep** sets. Then the ICD iteration concludes by orienting the resulting graph. Initially, v-structures are oriented and then FCI-orientation rules (Spirtes et al., 2000; Spirtes, 2001) are repeatedly applied until no more edges can be oriented. For completeness, in the last ICD iteration, a complete set of orientation rules is applied (Zhang, 2008).

The ICD algorithm is described in Algorithm 1. The main loop, lines 2–4, iterates over conditioning set sizes concurrently with the search radius on the graph. The iterative stage is described in function `Iteration`, lines 6–18. This function can be viewed as an operator that maps from an $(r-1)$-**O**-equivalence class to an $r$-**O**-equivalence. Thus, ICD is anytime in the sense that the main loop, lines 2–4, can be terminated for any value of $r$, resulting in a PAG that entails correct independence and causal relations. That is, terminating the loop after iteration $r = n$, results in a PAG that represents an $n$-**O-equivalence** class. Nevertheless, it still may not entail all relations that are entailed from the **O-equivalence** class of the underlying causal graph.

In the first ICD iteration, the initial PAG is a complete graph and $r = 0$. Thus every pair of nodes is tested for marginal independence (conditioning sets are empty). In the second iteration $r = 1$, where only nodes that are adjacent to the tested nodes are included in the conditioning set (PDS-path length limit is one edge). Only in succeeding iterations, nodes that are outside the neighborhood of the tested nodes may be included in the conditioning set. Conditioning sets, composed with from adjacent nodes or from outside the neighborhood, are returned by the function `PDSepRange`.

The result of `PDSepRange`$(X, Y, r, \mathcal{G})$, in Algorithm 1-line 9, is an ordered set of possible separating sets $\{\mathbf{Z}_i\}_{i=1}^{\ell}$, where each $\mathbf{Z}_i$ complies with the ICD-Sep conditions. The specific order in which these sets are used (Algorithm 1-line 12) to test conditional independence in Algorithm 1-line 13 may affect the total number of CI test in practice. One possible heuristic for ordering this set is such that $\mathbf{Z}_i$ sets are sorted based on to the average of the shortest PDS-path lengths connecting each node in $\mathbf{Z}_i$ to the tested nodes. First, for every set $\mathbf{Z} \in \{\mathbf{Z}_i\}_{i=1}^{\ell}$ created by `PDSepRange`, the following value is calculated,

$$\hat{d}_X(\mathbf{Z}) = \frac{1}{|\mathbf{Z}|} \sum_{W \in \mathbf{Z}} \min(|\Pi_Y(X, W)|), \tag{1}$$

where $\Pi_Y(X, W)$ is the PDS-path from $X$ to $W$, and $|\cdot|$ is path length. Then, the possible separating sets, $\mathbf{Z}_i$ are ordered according to this value. Note that the correctness of ICD is invariant to this order.

**Algorithm 1:** Iterative causal discovery (ICD algorithm)

---

**Input:**
    $n$: desired $n$-representing PAG (default: $|\mathbf{O}| - 2$)
    Ind: a conditional independence oracle

**Output:**
    $\mathcal{G}$: a PAG for $n$-**O**-equivalence class (a completed PAG is returned for the default $n = |\mathbf{O}| - 2$)

 

**1** initialize: $r \leftarrow 0$, $\mathcal{G} \leftarrow$ a complete graph with 'o' edge-marks, and $done \leftarrow$ False

**2** **while** $(r \leq n)$ & $(done = \text{False})$ **do**

**3**    $(\mathcal{G}, done) \leftarrow \texttt{Iteration}(\mathcal{G}, r)$  ▷ refine $\mathcal{G}$ using conditioning sets of size $r$

**4**    $r \leftarrow r + 1$

**5** **return** $\mathcal{G}$

 

**6** **Function** $\texttt{Iteration}(\mathcal{G}, r)$:

**7**    $done \leftarrow$ True

**8**    **for** *edge $(X, Y)$ in* $\text{edges}(\mathcal{G})$ **do**

**9**       $\{\mathbf{Z}_i\}_{i=1}^{\ell} \leftarrow \texttt{PDSepRange}(X, Y, \text{r}, \mathcal{G})$  ▷ $\mathbf{Z}_i$ complies with ICD-Sep conditions

**10**       **if** $\ell > 0$ **then**

**11**          $done \leftarrow$ False

**12**          **for** $i \leftarrow 1$ **to** $\ell$ **do**

**13**             **if** $\text{Ind}(X, Y | \mathbf{Z}_i)$ **then**

**14**                remove edge $(X, Y)$ from $\mathcal{G}$

**15**                record $\mathbf{Z}_i$ as a separating set for $(X, Y)$

**16**                **break**

**17**    orient edges in $\mathcal{G}$

**18**    **return** $(\mathcal{G}, done)$

---

### 3.3   An Example for the Difference between ICD and FCI

In this section we demonstrate the difference between the ICD and FCI recovering a simple graph that was used by Spirtes et al. (2000) to demonstrate FCI. In Figure 2, (a) is the true MAG, and (b) is its corresponding PAG. Both FCI and ICD start with an unoriented complete graph (absence of independence and causal information). In Figure 2 (c) the result of the first stage of FCI (PC skeleton and v-structures) is shown, and in Figure 2 (d) the result of ICD after iteration $r = 1$ (CI tests with up to one node in the conditioning set).

In both cases, the independence between $A$ and $E$ is not yet recovered, and both ICD and FCI require a similar number of CI tests with conditioning set of sizes 0 and 1. However, for concluding its first stage, FCI required additional 11 CI tests having conditioning set sizes 2 ($A$, $B$, $D$, and $E$, each has 3 neighbors: one indicates a tested edge while the other two serve as the conditioning set).

ICD continues after iteration $r = 1$ with increasing values of $r$ and recovers the independence between $A$ and $E$ after only 3 and 1 CI tests with conditioning set size of 2 and 3, respectively (only 4 conditioning sets comply with **ICD-Sep** conditions). The CI tests, with conditioning set sizes of 2 are $\text{Ind}(A, E | B, D)$, $\text{Ind}(A, E | B, F)$, $\text{Ind}(A, E | D, H)$ (no independence is found), and the single CI test having a conditioning set size of 3 is $\text{Ind}(A, E | B, D, F)$ (independence is found). At this point, ICD terminates. The second FCI stage (an iterative stage executed after concluding the first stage), requires an additional total of 76 CI tests with conditioning set sizes of up to 4 (76 conditioning sets comply with the definition of **Possible-D-Sep**). Overall, FCI requires 83 ($11 + 76 - 4$) additional CI tests compared to ICD. Note that for this specific example, ICD requires fewer CI tests than FCI's PC-stage alone.

(a) 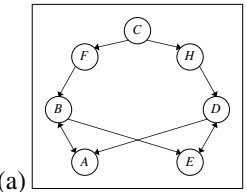 (b) 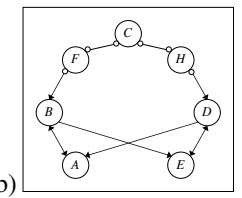 (c) 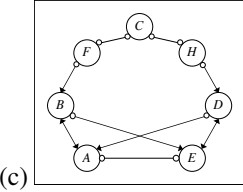 (d) 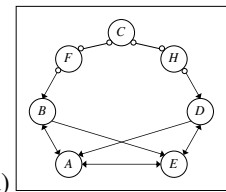

Figure 2: An example for comparing ICD with FCI using a simple 7-node graph that was used by Spirtes et al. (2000) to describe FCI. (a) A MAG corresponding to the true underlying DAG. (b) A PAG corresponding to the MAG. (c) Skeleton and v-structure orientations that is used by FCI (and FCI+). (d) The PAG resulting after ICD iteration $r = 1$. In both cases FCI (c) and ICD (d) it is desired to identify the independence between nodes $A$ and $E$. ICD requires significantly fewer CI tests compared to FCI (and PC in this specific case).

## 3.4 Correctness

We provide a sketch for the proof of correctness and completness of the ICD algorithm. The complete proof is in the supplementary material.

**Lemma 1.** *Let $\mathcal{G}$ be a PAG $n$-representing DAG $\mathcal{D}(\mathbf{O}, \mathbf{S}, \mathbf{L})$. Denote $A$, $B$ a pair of nodes from $\mathbf{O}$ that are connected in $\mathcal{G}$ and disconnected in $\mathcal{D}$, and such that $A$ is not an ancestor of $B$ in $\mathcal{D}$.*

*If $A \perp\!\!\!\perp B \mid [\mathbf{Z}'] \cup \mathbf{S}$, where $\mathbf{Z}' \subset \mathbf{O}$ is a minimal separating set having size $n + 1$, then there exists a subset $\mathbf{Z} \subset \mathbf{O}$ having the same size of $n + 1$ such that that $A \perp\!\!\!\perp B \mid \mathbf{Z} \cup \mathbf{S}$, and for every node $Z \in \mathbf{Z}$ there exists a PDS-path $\Pi_B(A, Z)$ in $\mathcal{G}$, such that every node $V$ on the PDS-path is also in $\mathbf{Z}$.*

In essence, from the definition of **D-Sep** (Spirtes et al., 2000, page 134 and Theorem 6.2), a node $\mathbf{Z}$ is in **D-Sep**$(A, B)$ if and only if in the MAG there is a path between $A$ and $V$ such that every node, except for the end points, is: 1. a collider and 2. an ancestor of $A$ or $B$ (an inducing path for $\rangle \mathbf{L}, \mathbf{S} \langle$). For every such path in a MAG, there exists a PDS-path in the corresponding PAG. In an $n$-representing PAG, we can rule out paths from being such a path in the MAG. Thus, for every such path in the MAG, there is a PDS-Path in an $n$-representing PAG, which ensures identifying at least one minimal separating set between ever pair of nodes that are m-separated in the MAG. In addition, every sub-path starting at $A$, of the PDS-path between $A$ and $V$, is also a PDS-path. This provides a link between the distance of the separating set nodes and the number of nodes in the separating set.

**Corollary 1.** *Let $\mathcal{G}$ be a PAG $n$-representing DAG $\mathcal{D}(\mathbf{O}, \mathbf{S}, \mathbf{L})$. Denote $A$, $B$ a pair of nodes from $\mathbf{O}$ that are connected in $\mathcal{G}$ and disconnected in $\mathcal{D}$.*

*If $A \perp\!\!\!\perp B \mid [\mathbf{Z}'] \cup \mathbf{S}$, where $\mathbf{Z}' \subset \mathbf{O}$ is a minimal separating set having size $n + 1$, then there exists a subset $\mathbf{Z} \subset \mathbf{O}$ having the same size of $n + 1$ such that that $A \perp\!\!\!\perp B \mid \mathbf{Z} \cup \mathbf{S}$, and for every node $Z \in \mathbf{Z}$ there exists a PDS-path $\Pi_B(A, Z)$ or $\Pi_A(B, Z)$, where every node $V$ on the PDS-path is also in $\mathbf{Z}$.*

The proof follows from Lemma 1.

**Lemma 2.** *Let $\mathcal{G}$ be a PAG $n$-representing a causal DAG $\mathcal{D}$. Let $\mathcal{S}$ be a skeleton (unoriented graph) that results after removing edges from the skeleton of $\mathcal{G}$ between every pair of nodes that are m-separated conditioned on a minimal separating set of size $n + 1$.*

*If $\mathcal{S}$ is oriented using anytime-FCI orientation rules, then the resulting graph is a PAG that $(n + 1)$-represents the causal DAG $\mathcal{D}$.*

The proof relies on the correctness of the anytime-FCI algorithm (Spirtes, 2001). This ensures the correctness of the orientation in each ICD-iteration.

**Proposition 1** (Correctness and completeness of the ICD algorithm). *Let $\mathcal{G}$ be a PAG representing a causal DAG $\mathcal{D}(\mathbf{O}, \mathbf{L}, \mathbf{S})$ and let $\mathrm{Ind}$ be a conditional independence oracle that returns d-separation relation for $\mathbf{O}$ in $\mathcal{D}$. If Algorithm 1 is called with $\mathrm{Ind}$, then the returned PAG, after uninterrupted termination is $\mathcal{G}$.*

We prove by mathematical induction. In each induction step $r + 1$ we prove using Corollary 1 that given an $r$-representing PAG, ICD iteration $r + 1$ finds all conditional independence relations having a minimal conditioning set of size $r + 1$, and removes corresponding edges. By Lemma 2, orientation

of the resulting graph results in an $(r + 1)$-representing PAG. Essentially, we prove that a minimal separating set complies with the ICD-Sep conditions, ensuring its identification in Algorithm 1-line 9.

### 3.5 Efficiency Analysis

We discuss the number of CI test required by ICD with respect to the number of observed variables $|\mathbf{O}|$ for learning an $n$-representing PAG ($n$-$\mathbf{O}$-equivalence). Namely, the complexity for returning a PAG after $n + 1$ iterations (recall that ICD is anytime). Let $\mathbb{D}^n$ be the class of causal DAGs for which the resulting PAG is also completed[2]. For all $\mathcal{D}(\mathbf{O}, \mathbf{S}, \mathbf{L}) \in \mathbb{D}^n$, $\forall A, B \in \mathbf{O}$, if $\exists \mathbf{Z} \subset \mathbf{O}$, such that $A \perp\!\!\!\perp B | \mathbf{Z} \cup \mathbf{S}$, then there exists a set $\mathbf{Z}' \subset \mathbf{O}$, such that $A \perp\!\!\!\perp B | \mathbf{Z}' \cup \mathbf{S}$ and $|\mathbf{Z}'| \leq n$. That is, its observable d-separation relations have at most $n$ nodes in their minimal separating sets. Nevertheless, for this class of DAGs, ICD may not terminate naturally after $n + 1$ iterations. ICD terminates naturally after $n + 1$ iterations (and returns a completed PAG) if in the true underlying PAG, $n$ is the size of the largest set complying with the ICD-Sep conditions. Consequently, $n$ is the largest conditioning set size considered by ICD.

The ICD algorithm starts with a complete graph and consists of a single loop, indexed by $r$. At iteration $r$, ICD considers in the worst case $\binom{|\mathbf{O}|}{2}$ edges, and for each edge, up to $2\binom{|\mathbf{O}|-2}{r}$ conditioning sets. Thus, the total number of CI tests is bounded by $N_{\max}$,

$$N_{\max} = 2\binom{|\mathbf{O}|}{2} \sum_{r=0}^{n} \binom{|\mathbf{O}| - 2}{r}. \tag{2}$$

In practice, the number of CI tests is significantly smaller. Firstly, up to iteration $r$ it is ensured that *all* the edges between nodes that are m-separated, in the true underlying MAG, by conditioning sets of sizes up to $r$ are removed. Secondly, the resulting PAG after each iteration is oriented using FCI orientation rules. These two operations reduce the sizes of PDS-trees in successive iterations, which leads to fewer **ICD-Sep** sets to consider as conditioning sets.

### 3.6 Stability

Constraint-based causal discovery algorithms rely on the accuracy of CI tests. In general, CI tests errors in early stages of an algorithm may lead to errors in later stage. For example, erroneously removing an edge in an early stage may lead to erroneously keeping an edge between nodes that are m-separated in the true underlying MAG, which in turn may lead to additional errors. Thus, in general it is desired that a causal discovery algorithm relies in early stages on statistical tests that have higher statistical power than statistical tests in later stages. Commonly, when limited data is available, statistical CI tests suffer from poor estimates of the statistic for large conditioning sets, compared to estimates for small conditioning sets. Thus, it is desired to use CI tests having small conditioning sets in early stages of the algorithm.

The FCI algorithm employs the PC algorithm (Spirtes et al., 2000) as an initial stage. The second stage of FCI relies on the accuracy of the resulting skeleton, where subsets of **Possible-D-Sep** are created based on this skeleton and used for further independence testing. The PC algorithm iterates over conditioning sets sizes and possibly concludes with CI tests having large conditioning sets. This might render the FCI algorithm unstable given limited database size.

The ICD algorithm benefits from a single iterative loop over conditioning set sizes (in contrast to FCI that has two loops that are executed consecutively). This ensures that CI with small conditioning set sizes are tested before CI test having larger conditioning sets. Thus, ICD is expected to be more stable than FCI and its related algorithms that have two iterative loops over the conditioning set sizes.

## 4 Experimental Evaluation

We evaluate the performance of ICD in terms of number of required CI tests and accuracy of the learned structures, and compare it to the performance of FCI (Spirtes et al., 2000), FCI+ (Claassen et al., 2013), and RFCI (Colombo et al., 2012).

---

[2]A completed PAG represents an equivalence class of MAGs such that no MAG can be ruled out given all CI relations. Not to be confused with a complete graph in which every node is connected to every other node.

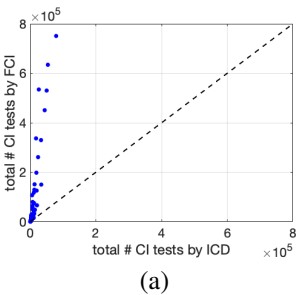 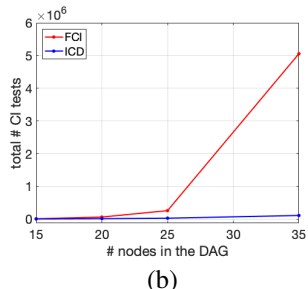

(a)                               (b)

Figure 3: Total number of CI tests. (a) A scatter plot using all DAGs in the experiment (ICD requires fewer CI tests than FCI for all the 100 tested DAGs). (b) Average total number of CI tests as a function of graph size.

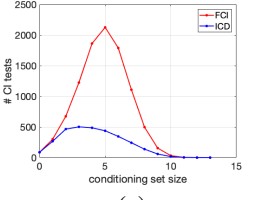 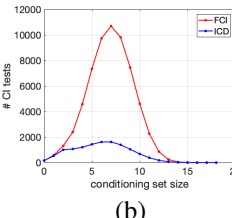 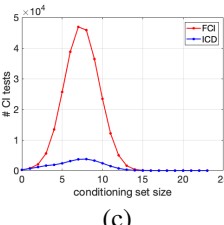 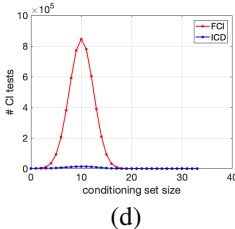

(a)              (b)              (c)              (d)

Figure 4: Average number of CI tests per conditioning set size for different graph sizes: (a) 15 nodes, (b) 20 nodes, (c) 25 nodes, (d) 35 nodes.

In all the experiments we follow a procedure, similar to the one described by Colombo et al. (2012) for generating random graphs with latent confounders. We create an adjacency matrix $\boldsymbol{A}$ for variables $\mathbf{O} \cup \mathbf{L}$ of DAG $\mathcal{D}(\mathbf{O}, \mathbf{L}, \mathbf{S} = \emptyset)$ by independent realization of $\mathrm{Bernoulli}(\rho/(n-1))$ in the upper triangle. If the resulting DAG is unconnected, we repeat until a connected DAG is sampled. For each DAG, we sample half of the parentless nodes that have at least two children and assign them to be the latent set $\mathbf{L}$ (making sure there is at least one). The remaining nodes are the observed set $\mathbf{O}$.

## 4.1 Number of Required CI Tests when using a Perfect CI Oracle

In the following experiments we evaluate the number of required CI tests by ICD, FCI, FCI+, and RFCI. We also analyze the conditioning set sizes of the required CI tests, since in many functions used for testing CI, the statistical power decreases and the computational complexity grows exponentially with the conditioning set size. For the following experiments in this section, we sample 100 random DAGs having $n \in \{15, 20, 25, 35\}$ nodes with a connectivity factor of $\rho = 2$ (25 DAGs per graph size). The same DAGs are used by each of the compared algorithms allowing a per-DAG comparison. A perfect CI oracle is implemented to returns d-separation relations in the true DAG.

### 4.1.1 ICD Compared to FCI

In this experiment we compare the ICD algorithm, using only the necessary ICD-Sep conditions (1 & 2), to the FCI algorithm—both are anytime, sound, and complete. ICD-Sep conditions 1 & 2 serve the key idea of ICD for constructing conditioning sets—*tying the condition set size to its distance from the tested nodes*.

From Figure 3 (a) it is evident that the ICD algorithm requires significantly fewer CI tests compared to FCI for all 100 tested graphs, and that this advantage of ICD is more dominant for graphs that require a larger number of CI tests. From Figure 3 (b) we find that the total number of CI tests required by ICD increases significantly more slowly with the graph size, compared to FCI. This difference is also evident in difference in run-times. We implemented FCI such that it uses the same routines as ICD, executed both algorithms on a single core of an Intel® Xeon® CPU, and measured runtime. The ratio FCI-runtime/ICD-runtime for graphs with 15, 20, 25, and 35 nodes is 1.3, 1.8, 2.9, and 5.6, respectively. As expected, this ratio increases with graph size.

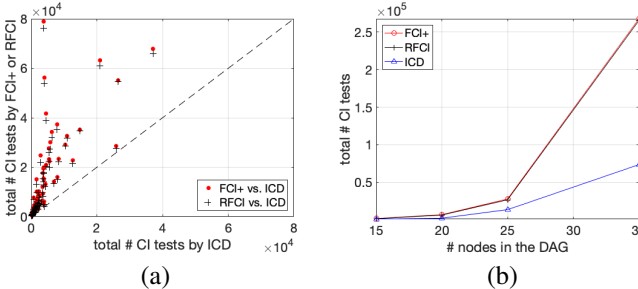

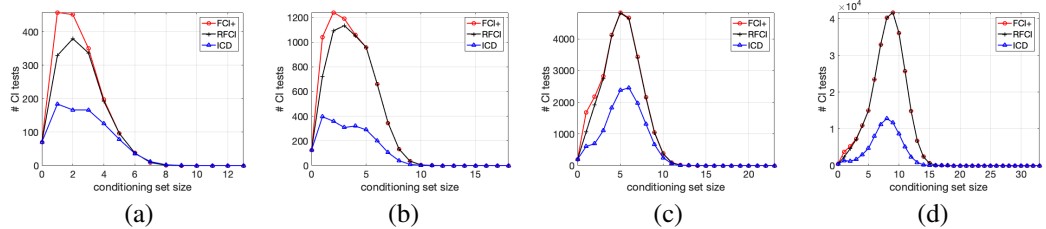

Figure 5: Total number of CI tests. (a) A scatter plot using all DAGs in the experiment (ICD requires fewer CI tests than FCI+ and RFCI for all the 100 tested DAGs). (b) Average total number of CI tests as a function of the graph size.

Figure 6: Average number of CI tests per conditioning set size for different graph sizes: (a) 15 nodes, (b) 20 nodes, (c) 25 nodes, (d) 35 nodes.

In Figure 4, for each tested graph size, we depict the average number of required CI tests per conditioning set size. There are three key observations, (1) ICD requires fewer CI tests than FCI for any conditioning set size, (2) the largest difference is evident for conditioning set sizes for which FCI required the most CI tests, and (3) this difference in the number of CI tests per conditioning set size increases with the graph size.

### 4.1.2 ICD Compared to FCI+ and RFCI

In this experiment we compare the ICD algorithm, using all ICD-Sep conditions, to the FCI+ and RFCI algorithms, which are improved versions of FCI, reducing the required number of CI tests. Both FCI+ and RFCI are sound. FCI+ is also complete, whereas RFCI is aimed at reducing the number of CI tests (compared to FCI) at the cost of not being complete. We were aided by the R package pcalg (Kalisch et al., 2012) for evaluating these algorithms.

In Figure 5 (a) it is demonstrated that ICD requires fewer CI tests than both FCI+ and RFCI[3]. On average, as evident from Figure 5 (b) the number of CI tests required by FCI+ and RFCI increases similarly with graph size, whereas this number for ICD grows significantly more slowly.

Lastly, we analyze the number of CI tests per conditioning set size, per graph size. Our observation from Figure 6 is threefold: (1) ICD requires fewer CI tests than FCI+ and RFCI for any conditioning set size, (2) the largest difference between ICD and FCI+/RFCI is evident for conditioning set sizes for which FCI+/RFCI, and (3) this difference increases with graph size, whereas the difference between FCI+ and RFCI becomes smaller relatively to the difference between them and ICD.

### 4.2 Structural Accuracy

In the following experiment we evaluate the accuracy of learned structures and the required number of statistical CI tests. To this end, we sample 100 DAGs, each having 15 nodes and an expected neighborhood 2, and quantify each edge of the DAGs by sampling from Uniform([-0.5, -2.0] [0.5, 2.0]). A probabilistic model is created by treating each node value as normally distributed with standard deviation 1, and mean being a weighted sum of the patents' values. From this model, for

---

[3]We found RFCI to require slightly fewer CI tests than FCI+ for all tested DAGs.

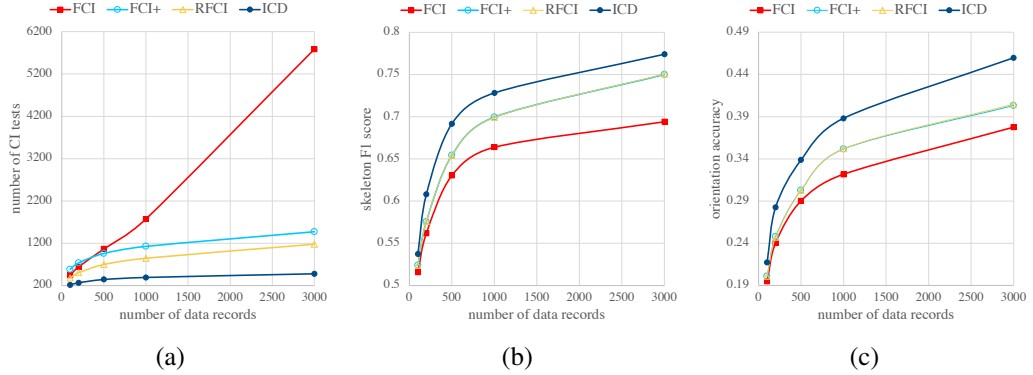

Figure 7: Average (a) number of CI tests, (b) accuracy of skeleton, and (c) accuracy of oriented edges, as a function of dataset size for the FCI, FCI+, RFCI, and ICD algorithms.

each of the 100 DAGs, we sample 5 data sets having sizes [100, 200, 500, 1000, 3000]. For each of the 500 data sets we learn graphical models using FCI, FCI+, RFCI, and ICD.

We measure the accuracy of the skeleton by calculating false-positive ratio (FPR), false-negative ratio (FNR) and F1-score. Correctly identifying the presence of an edge is considered true-positive. We measure the accuracy of edge orientation by calculating the percentage of correctly oriented edge-marks. Finally, we also count the number of CI tests required by each algorithm per data set. The average values (over 100 graphs) of the number of CI tests, skeleton F1 score, and orientation accuracy are summarized in Figure 7.

From the experiments, it is evident that ICD requires significantly fewer CI tests. Compared to the other methods, ICD has higher skeleton FPR (extra-edges), but lower skeleton FNR (missing edges, erroneously-identified independence relation). Overall, ICD has the highest F1 score. Lastly, it is evident that ICD has an advantage in orientation accuracy over the other methods.

## 5 Conclusions

We presented ICD, an anytime, sound, and complete causal discovery algorithm for learning PAGs representing $n$-**O-equivalence** classes. The ICD algorithm is a simple procedure that consists of a single loop over conditioning set sizes of CI tests. Having a single loop ensures that CI tests with small conditioning sets are tested before CI tests having larger conditioning set sizes. This can lead to greater stability in practical cases.

The ICD algorithm gradually increases the search radius, from a local neighborhood to the entire graph, for separating sets around connected nodes, resulting in an efficient search procedure. In early iterations, where the graph is dense and a small number of edges are oriented, the search for a separating set is localized. In later iterations, where the graph is sparser and more edges are oriented, a global search for a separating set becomes more efficient.

An important difference of the proposed ICD algorithm from FCI and its related algorithms is that, right from the outset it considers nodes for the conditioning set that are not in the local neighborhood of the tested nodes. One might suspect that this could result in a high number of CI tests evaluated by the ICD algorithm compared to the FCI algorithm. However when proceeding from one iteration to the next, the ICD algorithm reduces the number of nodes to consider for the conditioning sets by complete orientation in each iteration, and by limiting the distance of the conditioning nodes from the tested nodes.

Finally, from the experimental results, the ICD algorithm requires significantly fewer CI tests compared to FCI, and its related efficient algorithms FCI+ and RFCI, especially for large conditioning sets. Moreover, it is evident that the advantage of ICD increases with the graph size. In addition, ICD learns more accurate causal graphs. We believe that these advantages can be appealing to many real-world applications in domains such as economics, health, and social sciences.

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
