# Supplementary Material:
# Iterative Causal Discovery in the Possible Presence of Latent Confounders and Selection Bias

**Raanan Y. Rohekar**
Intel Labs
raanan.yehezkel@intel.com

**Shami Nisimov**
Intel Labs
shami.nisimov@intel.com

**Yaniv Gurwicz**
Intel Labs
yaniv.gurwicz@intel.com

**Gal Novik**
Intel Labs
gal.novik@intel.com

## A    Correctness and Completeness of the ICD Algorithm

In this section we provide a detailed proof for the correctness and completeness of the ICD algorithm. For easier referencing we describe ICD in Algorithm 1, and describe the ICD-Sep conditions. A set $\mathbf{Z}$ is a subset of $\mathbf{ICD\text{-}Sep}(A, B)$ given $r \in \{0, \dots, |\mathbf{O}| - 2\}$, if and only if

1. $|\mathbf{Z}| = r$,

2. $\forall Z \in \mathbf{Z}$, there exists a PDS-path $\Pi_B(A, Z)$ such that,

    (a) $|\Pi_B(A, Z)| \leq r$ and
    (b) every node on $\Pi_B(A, Z)$ is in $\mathbf{Z}$, and

3. $\forall Z \in \mathbf{Z}$, node $Z$ is a possible ancestor of $A$ or $B$ (not a necessary condition).

**Lemma 1.** *Let $\mathcal{G}$ be a PAG $n$-representing DAG $\mathcal{D}(\mathbf{O}, \mathbf{S}, \mathbf{L})$. Denote $A, B$ a pair of nodes from $\mathbf{O}$ that are connected in $\mathcal{G}$ and disconnected in $\mathcal{D}$, and such that $A$ is not an ancestor of $B$ in $\mathcal{D}$.*

*If $A \perp\!\!\!\perp B \mid [\mathbf{Z}'] \cup \mathbf{S}$, where $\mathbf{Z}' \subset \mathbf{O}$ is a minimal separating set having size $n + 1$, then there exists a subset $\mathbf{Z} \subset \mathbf{O}$ having the same size of $n + 1$ such that that $A \perp\!\!\!\perp B \mid \mathbf{Z} \cup \mathbf{S}$, and for every node $Z \in \mathbf{Z}$ there exists a PDS-path $\Pi_B(A, Z)$ in $\mathcal{G}$, such that every node $V$ on the PDS-path is also in $\mathbf{Z}$.*

*Proof.* It was previously shown that a minimal separating set for $A$ and $B$, where $A$ is not an ancestor of $B$, is a subset of $\mathbf{D\text{-}Sep}(A, B)$ (Spirtes et al., 2000, page 134 and Theorem 6.2; Spirtes et al., 1999). By definition, a node $Z$ is in $\mathbf{D\text{-}Sep}(A, B)$ if and only if in the MAG there is a path between $A$ and $Z$ such that every node, except for the end points, is: 1. a collider and 2. an ancestor of A or B. Denote such path DS-Path (an inducing path for $\langle \mathbf{L}, \mathbf{S} \rangle$). For every DS-Path in a MAG, there exists a PDS-path (possible-DS-path) in the corresponding PAG. In an $n$-representing PAG, we can rule out a path from being a DS-Path in the MAG if it contains at least one sub-path $X$ o–o $Y$ o–o $Z$, where $X$ and $Z$ are not connected, or if one of the nodes on the path (except the end points) is not a possible ancestor of $A$ or $B$. For an $n$-representing PAG, if a path between $A$ and $B$ has been ruled out of being a DS-Path, then identifying additional independence relations with conditioning set size greater than $n$ will not result in transforming this path into a DS-path. That is, $X$ o–o $Y$ o–o $Z$, where $X$ and $Z$ are not connected will not become an unshielded collider, and new ancestral relations identified in ICD-iteration $n + 1$ will not contradict ancestral relations identified in previous ICD-iterations (cf. Spirtes, 2001). Thus, for every DS-Path in the MAG, there is a PDS-Path in an $n$-representing PAG,

35th Conference on Neural Information Processing Systems (NeurIPS 2021).

consisting of the same sequence of nodes, which ensures identifying at least one minimal separating set between every pair $(A, B)$ that are m-separated in the MAG[1].

From the minimality of a separating set $\mathbf{Z}$ for $(A, B)$, $\forall Z \in \mathbf{Z}$ there is an open path between $Z$ and $A$ conditioned on $(\mathbf{Z} \setminus Z) \cup \mathbf{S}$, namely $A \not\perp\!\!\!\perp Z | (\mathbf{Z} \setminus Z) \cup \mathbf{S}$. Otherwise $Z$ is redundant and $\mathbf{Z}$ is not minimal. Let $\mathbf{C}$ be the set of nodes on a DS-path between $A$ and $Z$. A DS-path between $A$ and $Z$ is open (m-connected) conditioned on all the nodes between them on the path, $A \not\perp\!\!\!\perp Z | \mathbf{C} \cup \mathbf{S}$. For a PDS-path $\Pi_B(A, Z)$, $A \not\perp\!\!\!\perp Z | (\mathbf{Nodes}(\Pi_B(A, Z) \setminus \{A, Z\}) \cup \mathbf{S}$; namely, the PDS-path becomes an open path. Note that every sub-path starting from $A$ is also a PDS-Path. Thus, if $Z \in \mathbf{Z}$ then there exists a PDS-path connecting $A$ and $Z$ such that all the nodes on this path are in $\mathbf{Z}$. $\qquad\square$

**Corollary 1.** *Let $\mathcal{G}$ be a PAG $n$-representing DAG $\mathcal{D}(\mathbf{O}, \mathbf{S}, \mathbf{L})$. Denote $A, B$ a pair of nodes from $\mathbf{O}$ that are connected in $\mathcal{G}$ and disconnected in $\mathcal{D}$.*

*If $A \perp\!\!\!\perp B \mid [\mathbf{Z}'] \cup \mathbf{S}$, where $\mathbf{Z}' \subset \mathbf{O}$ is a minimal separating set having size $n + 1$, then there exists a subset $\mathbf{Z} \subset \mathbf{O}$ having the same size of $n + 1$ such that that $A \perp\!\!\!\perp B \mid \mathbf{Z} \cup \mathbf{S}$, and for every node $Z \in \mathbf{Z}$ there exists a PDS-path $\Pi_B(A, Z)$ or $\Pi_A(B, Z)$, where every node $V$ on the PDS-path is also in $\mathbf{Z}$.*

*Proof.* The proof follows from Lemma 1. Note that a minimal separating set for $A$ and $B$ is in $\mathbf{D}\text{-}\mathbf{Sep}(A, B)$ if $A$ is not an ancestor of $B$; otherwise, it is a subset of $\mathbf{D}\text{-}\mathbf{Sep}(B, A)$ if $B$ is not an ancestor of $A$. $\qquad\square$

**Lemma 2.** *Let $\mathcal{G}$ be a PAG $n$-representing a causal DAG $\mathcal{D}$. Let $\mathcal{S}$ be a skeleton (unoriented graph) that results after removing edges from the skeleton of $\mathcal{G}$ between every pair of nodes that are m-separated conditioned on a minimal separating set of size $n + 1$.*

*If $\mathcal{S}$ is oriented using anytime-FCI orientation rules, then the resulting graph is a PAG that $(n + 1)$-represents the causal DAG $\mathcal{D}$.*

*Proof.* We refer to the proof for the anytime FCI algorithm (Spirtes, 2001). It was shown, that a skeleton for any pair of disjoint nodes $A, B \in \mathbf{O}$, such that $A \perp\!\!\!\perp B | [\mathbf{Z}] \cup \mathbf{S}$ in $\mathcal{D}(\mathbf{O}, \mathbf{S}, \mathbf{L})$, where $\mathbf{Z} \subset \mathbf{O}$ and $|\mathbf{Z}| < n$, can be safely oriented by first orienting v-structures and then using the iterative FCI-orientation rules. Namely, it is sound in the sense that every orientation (head '—>' or tail '—') also exists in all that MAGs in the equivalence class of the true underlying MAG. Importantly, subsequent removal of edges, using conditioning set sizes greater than $n + 1$, will not invert the orientation of an edge-mark (a head will not be turned into a tail and vice versa), nor any oriented edge-mark (head or tail) will become invariant ('—o'). Zhang (2008) proved the completeness of the orientation rules step. Note that the proofs by Spirtes (2001) and Zhang (2008), both consider the presence of selection bias. $\qquad\square$

**Proposition 1** (Correctness and completeness of the ICD algorithm). *Let $\mathcal{G}$ be a PAG representing a causal DAG $\mathcal{D}(\mathbf{O}, \mathbf{L}, \mathbf{S})$ and let $\mathrm{Ind}$ be a conditional independence oracle that returns d-separation relation for $\mathbf{O}$ in $\mathcal{D}$. If Algorithm 1 is called with $\mathrm{Ind}$, then the returned PAG, after uninterrupted termination is $\mathcal{G}$.*

*Proof.* We prove by mathematical induction. In each induction step $r + 1$ we prove that given an $r$-representing PAG, ICD iteration $r + 1$ finds all conditional independence relations having a minimal conditioning set of size $r + 1$, and removes corresponding edges. By Lemma 2, orientation of the resulting graph results in an $(r+1)$-representing PAG. Essentially, we prove that a minimal separating set complies with the ICD-Sep conditions, ensuring its identification in Algorithm 1-line 9.

Let the true underlying DAG be $\mathcal{D}(\mathbf{O}, \mathbf{L}, \mathbf{S})$, and $\mathcal{G}$ be the graph returned after an ICD iteration. Throughout the proof $\mathbf{Z} \subset \mathbf{O}$, and $A \in \mathbf{O}, B \in \mathbf{O}$ are any pair of nodes.

***Base step ($r = 1$).*** The first ICD iteration $r = 0$ is trivial, where every pair of nodes is tested for marginal independence (ICD is initialized with a complete graph). From Lemma 2, the orientation of the graph using FCI-orientation rules returns a 0-representing PAG. We define our base case for the second ICD iteration $r = 1$. Minimal separating set consisting of a single node are sought. Let $A \perp\!\!\!\perp B | [\mathbf{Z}] \cup \mathbf{S}$ in $\mathcal{D}$, such that $|\mathbf{Z}| = 1$ (a single-node set). The ICD-Sep conditions for $r = 1$ effectively restrict the search to the neighborhood of $A$ and $B$. Although there may be multiple

---

[1]Spirtes et al. (2000) defined **Possible-D-Sep** as a super-set of **D-Sep** based on the PDS-path generalization of DS-paths.

separating single-node sets for $(A, B)$, there exist at least one in their neighborhood. Recall that by conditioning on a separating set, paths between $A$ and $B$ are blocked. Since we are considering single-node separating sets, there exists an active path that is blocked by a single node, such that it does not consist any collider (otherwise the collider is included in the separating set and the size is greater than 1). Thus, this path can be blocked by at least one of the neighbors of $A$ and $B$. This ensures that considering only neighbors of the tested nodes, all the independence relations with minimal separating sets of size one are identified, and corresponding edges are removed (Algorithm 1-lines 12–16). Following Lemma 2, orientation using FCI-orientation rules ensures that the resulting graph is a PAG that 1-represents the causal DAG $\mathcal{D}$.

***Induction step $(r + 1)$.*** Let $A \perp\!\!\!\perp B | [\mathbf{Z}] \cup \mathbf{S}$ in $\mathcal{D}$, such that $|\mathbf{Z}| = r + 1$. From Corollary 1, there exists a separating set of size $r + 1$ that complies with ICD-Sep condition 2. Condition 1 is complied by definition. This ensures identifying all independence relations with a minimal separation set of size $r + 1$ are identified and corresponding edges are removed (Algorithm 1-lines 12–16). Following Lemma 2, orientation using FCI-orientation rules ensures that the resulting graph is a PAG that $(r + 1)$-represents the causal DAG $\mathcal{D}$.

From the definition of **ICD-Sep** it follows that for a pair of adjacent nodes $A$ and $B$, if **ICD-Sep**$(A, B)$ given $r$ is empty, then **ICD-Sep**$(A, B)$ given $r + 1$ is empty. Thus, concluding the algorithm at iteration $r$ if $|\textbf{ICD-Sep}(A, B)|$ is empty for any ordered pair of adjacent nodes $(A, B)$ ensures that all independence relations have been identified, which ensures completeness. $\square$

---

**Algorithm 1:** Iterative causal discovery (ICD algorithm)

---

**Input:**
    $n$: desired $n$-representing PAG (default: $|\mathbf{O}| - 2$)
    Ind: a conditional independence oracle

**Output:**
    $\mathcal{G}$: a PAG for $n$-**O**-equivalence class (a completed PAG is returned for the default $n = |\mathbf{O}| - 2$)

---

1   initialize: $r \leftarrow 0$, $\mathcal{G} \leftarrow$ a complete graph with 'o' edge-marks, and $done \leftarrow$ False

2   **while** $(r \leq n)$ & $(done = \text{False})$ **do**
3     $\quad (\mathcal{G}, done) \leftarrow \texttt{Iteration}(\mathcal{G}, r)$   ▷ `refine` $\mathcal{G}$ `using conditioning sets of size` $r$
4     $\quad r \leftarrow r + 1$

5   **return** $\mathcal{G}$

6   **Function** $\texttt{Iteration}(\mathcal{G}, r)$**:**
7     $done \leftarrow$ True
8     **for** *edge* $(X, Y)$ *in* $\text{edges}(\mathcal{G})$ **do**
9        $\{\mathbf{Z}_i\}_{i=1}^{\ell} \leftarrow \texttt{PDSepRange}(X, Y, r, \mathcal{G})$   ▷ $\mathbf{Z}_i$ `complies with ICD-Sep conditions`
10       **if** $\ell > 0$ **then**
11          $done \leftarrow$ False
12          **for** $i \leftarrow 1$ **to** $\ell$ **do**
13             **if** $\text{Ind}(X, Y | \mathbf{Z}_i)$ **then**
14                remove edge $(X, Y)$ from $\mathcal{G}$
15                record $\mathbf{Z}_i$ as a separating set for $(X, Y)$
16                **break**

17     orient edges in $\mathcal{G}$
18     **return** $(\mathcal{G}, done)$

---

# B    Additional Experimental Results for Structural Accuracy

In this section we provide Table 1, the experimental results discussed in Section 4.2 of the paper.

We measure the accuracy of the skeleton by calculating false-positive ratio (FPR), false-negative ratio (FNR) and F1-score. Correctly identifying the presence of an edge is considered true-positive.

We measure the accuracy of edge orientation by calculating the percentage of correctly oriented edge-marks. Finally, we also count the number of CI tests required by each algorithm and normalize it by the number required by FCI (per data set).

Table 1: Structural accuracy of learned graphs and the required number of CI tests for FCI, FCI+, RFCI, and (proposed) ICD. The accuracy of graph skeleton is measured by false-positive ratio (FPR), false-negative ratio (FNR), and F1 score. Edge orientation accuracy is measured by the percentage of correctly oriented edges.

| Data Samples | Algorithm | # CI Tests Ratio | FPR | FNR | F1 Score | Orientation Accuracy |
|---|---|---|---|---|---|---|
| 100 | FCI | 1.0 | **0.010** | 0.638 | 0.52 | 0.19 |
| 100 | FCI+ | 1.3 | 0.012 | 0.628 | 0.52 | 0.20 |
| 100 | RFCI | 0.9 | 0.012 | 0.628 | 0.52 | 0.20 |
| 100 | ICD | **0.5** | 0.023 | **0.606** | **0.54** | **0.22** |
| 200 | FCI | 1.0 | **0.014** | 0.589 | 0.56 | 0.24 |
| 200 | FCI+ | 1.1 | 0.019 | 0.570 | 0.58 | 0.25 |
| 200 | RFCI | 0.8 | 0.019 | 0.570 | 0.58 | 0.25 |
| 200 | ICD | **0.4** | 0.039 | **0.515** | **0.61** | **0.28** |
| 500 | FCI | 1.0 | **0.011** | 0.519 | 0.63 | 0.29 |
| 500 | FCI+ | 0.9 | 0.017 | 0.485 | 0.65 | 0.30 |
| 500 | RFCI | 0.7 | 0.016 | 0.485 | 0.65 | 0.30 |
| 500 | ICD | **0.3** | 0.062 | **0.389** | **0.69** | **0.34** |
| 1000 | FCI | 1.0 | **0.008** | 0.482 | 0.66 | 0.32 |
| 1000 | FCI+ | 0.6 | 0.018 | 0.428 | 0.70 | 0.35 |
| 1000 | RFCI | 0.5 | 0.017 | 0.429 | 0.70 | 0.35 |
| 1000 | ICD | **0.2** | 0.081 | **0.320** | **0.73** | **0.39** |
| 3000 | FCI | 1.0 | **0.005** | 0.447 | 0.69 | 0.38 |
| 3000 | FCI+ | 0.25 | 0.020 | 0.359 | 0.75 | 0.40 |
| 3000 | RFCI | 0.20 | 0.019 | 0.360 | 0.75 | 0.40 |
| 3000 | ICD | **0.08** | 0.111 | **0.209** | **0.77** | **0.46** |

# C   Broader Impact

The significant progress made in ML research over the past few years, has led to increasing deployment of algorithms in real world applications. While state of the art models often reach high quality results, they have been criticized for making black box decisions, not providing their users tools to explain how they reach their conclusions. Understanding how models arrive at their decisions is critical for the use of AI, as it builds users' trust in ML based automatic systems, especially in decision critical applications. Such trust can be built by giving the user insights on how a system reaches its conclusions, which is especially important with high dimensional data having a large number of domain variables to consider. Causal structure discovery provides various capabilities beyond inference, such as counterfactual analysis, association, intervention, and imagining, and therefore, may also serve as an addition to the Explainable AI toolset, as it aims to improve the ability to identify the causal relationships among domain variables, thereby providing the decision makers with tools to understand those decisions, e.g. which variables are important and to what degree? which do not influence a specific result? which domain variables are the cause of a phenomena, and which merely correlate with it? By having such ability, human operators can supervise the recommendations of the method, intervene and point to cases that, in their view as experts, are potentially arguable, and therefore require an in-depth analysis before concluding with a final decision. Positive examples of such are abundant, especially from observational clinical data, and offer guidance to accurately discover known causal relationships in the medical domain. Fairness, inequality and bias issues, e.g. against minorities, oftentimes exist in data, and our approach, through the causal graph, provides the human supervisor with an inherent ability to inquire the ruling of the algorithm, thereby to consider potential ethical bridges that may reside in the causal graph, and consequently to overrule and correct them. With this innate transparency, we believe that our method is posited better to handle some of those ethical concerns, reinforcing the users' trust in the method, and positively impacting their willingness to use and rely on it.