# OpenReview forum: "Iterative Causal Discovery in the Possible Presence of Latent Confounders and Selection Bias"
_NeurIPS.cc/2021/Conference — NeurIPS 2021 Poster_

### Official Review · Reviewer_v9pH · 2021-07-05

**Rating:** 6
**Confidence:** 3

**Summary:**

The paper proposes a method for causal discovery in graphs with potential latent common causes, which compared to other similar methods requires significantly less conditional independence tests and smaller conditioning sets. It achieves that by incrementally increasing the size of the conditioning set relative to the distance from the tested nodes. This approach bases the incrementally learnt graph on statistical tests with minimum size of conditioning set, therefore theoretically boosting its statistical strength and stability. Without additional assumptions the method is able to detect PAGs.

**Ethical Concerns:**

No ethical concern.

**Limitations And Societal Impact:**

The main limitation of the paper, in my opinion, is that it does not provide any experimental results (neither on simulated nor on real data) about the false positive and false negatives of the method. I would request the authors to do so, comparing the accuracy of their incremental method with the existing methods, with respect to FPR, FNR.

Other points:
- Proposition 1 is not well written. The G that the algorithm will return depends on the 'n' number of iterations.
-Line 61: Do you mean returning every time a correct *subgraph* instead of graph? It cannot be that at any step of the iteration the algorithm returns the correct graph since it has not examined all the nodes.
-Line 73: A CI test is commonly a statistical hypothesis test used to determine from observed data whether two variables are correlated conditioned on a set of some other variables. -> Not just correlated. Whether they are dependent in general (not just linear dependency).
-Could you elaborate on the statement on line 126?
-The authors do not provide any paragraph for the Broader Impact of their paper.

**Main Review:**

The authors provide theoretical proofs of their method and experimental comparisons with FCI and its variants. Nevertheless, the authors report results only in terms of numbers of CI tests and size of conditioning set, rather than false positive and false negative rates. As a result I cannot judge how well their algorithm performs in terms of discovery of the graph. From the theory point of view, their algorithm is both efficient and stable, as it is built on CI tests with small conditioning sets. This is also exhibited in the experiments.

If the authors provide performance - related experiments (FPR, FNR) in comparison with the FCI methods, in order to see whether the algorithm works in practice I am willing to increase my grade.

**Time Spent Reviewing:**

10 hours

---

> ### Author Response · Authors · 2021-08-11
> **Structural accuracy in terms of FPR, FNR, F1 measure, and edge orientation accuracy**
>
> We sincerely appreciate your request for additional evaluation of ICD using a statistical independence test for CI testing.
>
> **Setup.** We sampled 100 DAGs, each having 15 nodes and expected neighborhood 2. We randomly sampled half of the parentless nodes that have at least two children, and marked them as hidden confounders (making sure there is at least one). We quantified each edge of the DAG by sampling from Uniform([-0.5, -2.0] $\cup$ [0.5, 2.0]). A probabilistic model is created by treating each node value as normally distributed with standard deviation 1, and mean being a weighted sum of the patents' values. From this model, for each of the 100 DAGs, we sampled 4 datasets having sizes [100, 200, 500, 1000].
>
> **Evaluation.** As requested, we evaluate FPR and FNR for each of the algorithms: FCI, FCI+, RFCI, and ICD. Correctly identifying the presence of an edge is considered true-positive. We also evaluate the number of CI tests required by each algorithm and normalize it by the number required by FCI (under the same setting). The average number of CI tests required by FCI for dataset sizes 100, 200, 500, and 1000 is 246, 531, 1234, and 2800, respectively. Finally, we evaluate F1 score, and percentage of correctly oriented edges (edges present in both skeletons of the true and learned graphs). The average values (over 100 graphs) are summarized in the following table.
>
> **Results.** From the experiments, ICD requires significantly fewer CI tests. It has higher FPR (extra-edges), but lower FNR (missing edges, erroneous independence relation). In addition, ICD has an advantage in orientation accuracy and F1 score.
>
> *Replies to other points*
>
> + We will add to Proposition 1 the condition of terminating ICD (prematurely) at iteration *n*. The proof remains valid (see a more detailed proof, by induction, in the appendix).
> + In line 61 *iteration* means an evaluation of possible separating sets within a given search radius (lines 59-60) for all connected nodes. We will rephrase lines 59-60 for clarifying the meaning of *iteration*.
> + We will replace the *correlation* term with *statistically dependent*.
> + By definition ICD-Sep conditions include the conditions for Possible-D-Sep (set size = *r* and the existence of a PDS-path). Since ICD-Sep condition includes additional conditions (2.a, 2.b, and 3), {$S | S\subset \mathbf{DSep}, |S|=r$} $\subset$ $\mathbf{ICDSep}_{r}$ $\subset$ {$S | S\subset \mathbf{PossibleDSep}, |S|=r$}. We will include this clarification in the text.
> + We will include a broader impact discussion in the paper (requirement 1-c of the checklist). Please see the discussion detailed in our response to reviewer RqyE.
>
>
>
>
> | data size | algorithm | # CI tests ratio |  FPR  |  FNR  |  F1 score  | Orientation Accuracy |
> |:---------:|-----------|:----------------:|:-----:|:-----:|:----:|:---------------------:|
> |    100    | FCI       |        1.0       | **0.003** | 0.534 | 0.62 |          0.31         |
> |    100    | FCI+      |        1.9       | 0.004 | 0.504 | 0.65 |          0.24         |
> |    100    | RFCI      |        1.1       | 0.004 | 0.504 | 0.65 |          0.24         |
> |    100    | ICD       |        **0.7**     | 0.009 | **0.482** | **0.66** |          **0.33**         |
> |           |           |                  |       |       |      |                       |
> |    200    | FCI       |        1.0       | **0.003** | 0.449 |  0.7 |          0.38         |
> |    200    | FCI+      |        1.1       | 0.007 | 0.405 | 0.73 |          0.24         |
> |    200    | RFCI      |        0.6       | 0.007 | 0.405 | 0.73 |          0.24         |
> |    200    | ICD       |        **0.4**     | 0.014 | **0.369** | **0.74** |          **0.41**         |
> |           |           |                  |       |       |      |                       |
> |    500    | FCI       |        1.0       | **0.002** | 0.374 | 0.75 |          0.47         |
> |    500    | FCI+      |        0.6       | 0.007 | 0.313 |  0.8 |          0.25         |
> |    500    | RFCI      |        0.4       | 0.006 | 0.313 |  0.8 |          0.25         |
> |    500    | ICD       |        **0.2**     | 0.022 | **0.268** | **0.81** |          **0.51**         |
> |           |           |                  |       |       |      |                       |
> |    1000   | FCI       |        1.0       | **0.001** | 0.323 |  0.8 |          0.52         |
> |    1000   | FCI+      |        0.3       | 0.006 | 0.255 | 0.84 |          0.25         |
> |    1000   | RFCI      |        0.2       | 0.006 | 0.255 | 0.84 |          0.25         |
> |    1000   | ICD       |        **0.1**     | 0.028 | **0.198** | **0.85** |          **0.58**         |

---

### Official Review · Reviewer_RqyE · 2021-07-14

**Rating:** 7
**Confidence:** 2

**Summary:**

The authors propose an algorithm for constraint-based causal discovery in the presence of latent confounders and selection bias.
It is similar to FCI while it can have better efficiency than FCI in practice with fewer conditional independence tests. The paper also included sufficient analysis of the algorithm and the experimental results support the claims.

**Ethics Review Area:**

["I don’t know"]

**Limitations And Societal Impact:**

As for limitations, yes; As for social impact, it showed in the checklist that "N/A" in 1(c) of the checklist. Maybe some general concern of causal discovery algorithms, such as the impact of potential errors because of wrongly using the methods?

**Main Review:**

The algorithm contributes to the constraint-based methods in the presence of latent confounders and selections bias from the efficiency perspective. Based on the experimental results, efficiency analysis, and stability analysis, the algorithm can work as the author claimed. The experiments are properly done and the comparison with FCI is clear. So the contribution is sound and it can be an alternative to FCI in many cases.

**Time Spent Reviewing:**

2

---

> ### Author Response · Authors · 2021-08-11
> **Broader impact**
>
> Thank you for your review. Upon your request regarding potential negative impact, we will include in the introduction a discussion regarding the potential benefits and challenges when applying causal reasoning algorithms using learned graphs. Specifically, we will embed the following paragraph in the paper.
>
> >The significant progress made in ML research over the past few years, has led to increasing deployment
> of algorithms in real world applications. While state of the art models often reach high quality results,
> they have been criticized for making black box decisions, not providing their users tools to explain
> how they reach their conclusions. Understanding how models arrive at their decisions is critical
> for the use of AI, as it builds users’ trust in ML based automatic systems, especially in decision critical applications.
> Such trust can be built by giving the user insights on how a system reaches
> its conclusions, which is especially important with high dimensional data having a large number
> of domain variables to consider. Causal structure discovery provides various capabilities beyond
> inference, such as counterfactual analysis, association, intervention, and imagining, and therefore,
> may also serve as an addition to the Explainable AI toolset, as it aims to improve the ability to identify
> the causal relationships among domain variables, thereby providing the decision makers with tools to
> understand those decisions, e.g. which variables are important and to what degree? which do not
> influence a specific result? which domain variables are the cause of a phenomena, and which merely
> correlate with it? By having such ability, human operators can supervise the recommendations of
> the method, intervene and point to cases that, in their view as experts, are potentially arguable, and
> therefore require an in-depth analysis before concluding with a final decision. Positive examples
> of such are abundant, especially from observational clinical data, and offer guidance to accurately
> discover known causal relationships in the medical domain. Fairness, inequality and bias issues, e.g.
> against minorities, oftentimes exist in data, and our approach, through the causal graph, provides the
> human supervisor with an inherent ability to inquire the ruling of the algorithm, thereby to consider
> potential ethical bridges that may reside in the causal graph, and consequently to overrule and correct
> them. With this innate transparency, we believe that our method is posited better to handle some
> of those ethical concerns, reinforcing the users’ trust in the method, and positively impacting their
> willingness to use and rely on it.

---

> > ### Comment · Reviewer_RqyE · 2021-09-01
> > **Response**
> >
> > Thank the authors for providing the negative impact and discussion. It looks pretty good.

---

### Official Review · Reviewer_79JW · 2021-07-16

**Rating:** 6
**Confidence:** 4

**Summary:**

This paper presents a variant of the FCI algorithm for causal discovery in the presence of latent confounding and selection bias. The proposed algorithm is claimed to be anytime, sound and complete, and is shown to typically require significantly fewer conditional independence tests than other variants of FCI.

**Limitations And Societal Impact:**

Adequately addressed.

**Main Review:**

The idea of this paper is interesting and promising. If the claims are true, the contribution is in my view fairly significant, as FCI remains an important algorithm to tackle causal discovery when latent confounding and selection bias cannot be assumed away and the large number of conditional independence tests needed in FCI is a major concern. However, I am not yet convinced of the claims of the paper. My main concerns are the following:

1. I do not follow the proof of Lemma 1. First of all, the statement of the lemma is not clear, as the quantifier in front of "Z\in \mathbf{Z}" is not specified. Given the definition of ICD -Sep on p. 3, I gather that a universal quantifier is intended here. If so, I suspect the lemma, as stated, is false. For example, suppose the graph is A --> X <-- Y --> B and X --> Z --> W --> B. Now A and B are d-separated by {Y, Z}, which is a minimal separating set. But Z does not satisfy the condition stated in the lemma. In this case, there is a minimal separating set, i.e., {Y, X}, which satisfies the condition in question, so I guess the intended lemma here should be IF A and B are d-separated by a set with size n+1, THEN there exists a separating set of that size that satisfies the condition. In any case, the proof does not manage to establish that, as far as I can see.

2. The proof of Proposition 1 cites Spirtes's (2001) proof that FCI-orientation rules are still applicable even if the skeleton contains extra edges. However, this paper claims to also apply Zhang's (2008) orientation rules, and it is not clear that the rules considered by Spirtes are sufficient to cover all the rules in the latter.

3. It is also not entirely clear what a PAG n-representing the underlying DAG needs to be. Spirtes's (2001) definition of a PAG does not require orientation completeness, but this paper seems to require that, as indicated on p. 2. What it means for a PAG n-representing the underlying DAG to be complete should be clearly defined, and that such a PAG would be returned by the algorithm when it is interrupted after the n'th round needs to be more carefully demonstrated.


Minor comments:

1. The definition of "possible ancestor" (Definition 4) is a little confusing. Does it require only the first edge on the path to satisfy the said condition?

2. On p. 4, it is claimed that ICD-Sep is a (smaller) super-set of D-Sep. This is confusing on two counts. First, ICD-Sep is said to be
"sets complying with ICD-Sep conditions", which seems to suggest that it is a set of sets of nodes, whereas D-Sep is a set of nodes. Second, ICD-Sep has a parameter r, whereas D-Sep does not, at least as it is defined in the paper.



**Time Spent Reviewing:**

5

---

> ### Author Response · Authors · 2021-08-11
> **Initial response**
>
> *We are sincerely thankful for your detailed insights on the theoretical results.*
>
> 1. Due to lack of space in the main paper, we provided a detailed proof in the appendix (supplementary material), where Lemma 1 in the paper is Corollary 1 in the appendix. We thank you for your feedback on the current proof and the statement in Lemma 1. We will update them based on your suggestions, and by including additional details from the appendix.
>     - *Regarding the possible counter-example you provided*: There are three minimal sets that separate A and B: {X,Y} in D-Sep(A,B) and ICD-Sep(A,B), {W,Y} in D-Sep(B,A) and ICD-Sep(B,A), and {Y,Z} in ICD-Sep(B,A) in the 1-representing PAG. Lemma 1 requires an *n*-representing PAG before referring to separating sets of size *n*+1. In the example, *n*=1 (since we seek a size 2 separating set for A and B). Note that B and Z are connected in such a PAG (1-representing), as the minimal separating set for them has size 2, {W,Y} (or {W,X}). Thus, Z and Y are in ICD-Sep(B,A), since they are neighbors of B.
>     Although this example does not contradict Lemma 1, we will rephrase it as you (and reviewer WByU) suggested.
>     - We will rephrase Lemma 1 such that it states that we are referring to *at least one* minimal set (if there are several).
>     - We will add details from the appendix to the proof sketch. In essence, from the definition of D-Sep (Spirtes et al. 2000, page 134), a node *Z* is in D-Sep(A,B) if and only if in the MAG there is a path between A and V such that every node, except for the end points, is: 1. a collider and 2. an ancestor of A or B. Denote such path DS-Path (an inducing path for <L,S>). For every DS-Path in a MAG, there exists a PDS-path in the corresponding PAG. In an *n*-representing PAG, we can rule out paths from being a DS-Path in the MAG if it contains at least one sub-path X o--o Y o--o Z, where X and Z are not connected, or if one of the nodes on the path (except the end points) is not a possible ancestor of A or B.
>     For an *n*-representing PAG, if a path between A and B has been ruled out of being a DS-Path, then identifying additional independence relations with conditioning set size greater than *n* will not result in transforming this path into a DS-path. That is, X o--o Y o--o Z, where X and Z are not connected will not become an unshielded collider, and new ancestral relations identified in ICD-iteration n+1 will not contradict ancestral relations identified in previous ICD-iterations (Spirtes 2001).   Thus, for every DS-Path in the MAG, there is a PDS-Path in an *n*-representing PAG, which ensures identifying at least one minimal separating set between ever pair (A,B) that are m-separated in the MAG.
>
> 2. *Regarding the use of orienting rules in Algorithm 1*: your observation is accurate. We will clarify the distinction between the orientation rules by Spirtes (2001) and the complete set of rules by Zhang (2008b). In Algorithm 1-line 17 the graph is oriented by rules of Spirtes (2001). The complete set of rules by Zhang (2008b) is applied once ICD concludes (the last iteration). All the experiments in the paper use this scheme.
> 3. We follow the definition of Spirtes (2001) for an *n*-representing PAG.
> Since the orientation rules are driven by the identified conditional independence, orientation completeness is ensured only after identifying all observed conditional independence relations. Thus, orientation completeness is ensured by ICD only after a natural termination (without interruption). We will clarify this point.
>
>
> *Replies to minor comments*
>
> 1. There is a typo in Definition 4. The edge should be indexed with *j* instead of *i*, for all *j* in {*i*,...,*i*+*k*-1}.
> 2. We will clarify that $\mathbf{ICD-Sep}$ with parameter *r* is a smaller super-set of {$S | S\subset \mathbf{DSep}, |S|=r$}. That is, {$S | S\subset \mathbf{DSep}, |S|=r$} $\subset$ $\mathbf{ICDSep}_{r}$ $\subset$ {$S | S\subset \mathbf{PossibleDSep}, |S|=r$}.

---

> > ### Comment · Reviewer_79JW · 2021-08-31
> > **Score increased**
> >
> > Thanks for the helpful response. I am now leaning towards a positive recommendation.

---

### Official Review · Reviewer_WByU · 2021-07-21

**Rating:** 6
**Confidence:** 3

**Summary:**

The authors describe an algorithm for learning causal relations in the presence of latents and selection bias. The algorithm called ICD recovers the partial ancestral graph (PAG) corresponding to "some" equivalence class of the maximal ancestral graph of the underlying DAG. Most prior work for this problem studied the problem of recovering PAG using FCI variants.

In this work, the authors outline a greedy neighborhood search-based algorithm (ICD) that can be stopped after "r" iterations, and the PAG recovered by ICD corresponds to an "r-equivalence class". By definition, an "r-equivalence class" encodes all causal relations that can be recovered using conditional independence tests that use at most "r" observable nodes. Therefore, the guarantees of this paper are weaker than the recovery guarantees of FCI which return the PAG of the equivalence class. Note that FCI uses conditioning sets that can be of size as large as the size of the total observable nodes.

FCI is an anytime algorithm and can be stopped after "r" iterations and outputs an "r-equivalence class". In the first stage of FCI, the PC algorithm returns a set of nodes S and a partial graph. Then, FCI checks subsets of S in the order of increasing size for conditional independencies. Finally, the orientation rules that are sound and complete are applied. In this paper, the authors suggest a way of bypassing this step-wise procedure, directly generating subsets of increasing size and checking for conditional independencies. Finally, they suggest using the orientation rules of FCI (when asked to terminate after "r"th iteration). Although there is no theoretical improvement in the number of conditional independence tests used, the authors empirically demonstrate that their approach is better than FCI in the number of conditional independence tests used.

**Main Review:**


The problem that the authors' study is an important problem, and in general the authors take an interesting approach to solve it. I find that the algorithm is a natural idea of greedily growing the neighborhoods using the existing paths, and ensuring that the separating sets are identified correctly (ICD-Sep conditions in the paper). Although the algorithm is not entirely novel, the soundness and completeness of the algorithm make it a good contribution.

I would request the authors to clarify a few things:

1. How "far away" is the n-equivalence from the total equivalence, i.e., |O|-equivalence? I am not fully convinced why the notion of this specific equivalence class is useful.

2. Lemma 1 -- why does there exist a PDS path Pi(A, Z) for every Z? I understand if one exists, your proof follows.

3. In general, it would be interesting to understand if there is an actual improvement in the conditional independence tests (theoretically), even for certain graph families.

4. (Lines 285-288) What is the actual time it took in the implementation of your algorithms?

5. Figure 4 on Page 8? How are you able to change the conditioning set size of FCI -- isn't it determined by the PC algorithm in the first stage?

Other comments:

-- Please state the problem of "anytime learning of causal graphs" formally in the preliminaries or introduction.

-- (Line 80) Defn 1. For "every possible" X, Y, Z..

-- (Line 95) What is a subpath?

-- Line 17 of Algorithm 1. Orient edges "using ..." specify the rules/idea fully.

-- Try relating the computational complexity based on the diameter of the graphs. The constant diameter or low diameter graphs are known to capture many graph families.

-- Please add a reference to the paper "Learning Causal Graphs with Small Interventions", and differentiate the notion of separating sets from this to yours.

**Time Spent Reviewing:**

15-20

---

> ### Author Response · Authors · 2021-08-11
> **Initial response**
>
> *We truly appreciate your detailed review and suggestions.*
>
> 1. Upon your request, we will include a definition of $n$-O-equivalence class in the paper. The definition of $n$-O-equivalence class was defined by Spirtes (2001). It is used in the paper to describe the result of ICD if prematurely terminated after iteration $n$.
> 2. Regarding Lemma 1, you are correct. We will rephrase Lemma 1 and clarify that a PDS-path exists for at least one minimal separating set.
> 3. Thank you for suggesting to relate the complexity to the graph diameter. However, the diameter of the graph skeleton considers shortest paths between nodes, whereas PDS-paths may not be the shortest paths. When considering the longest paths instead, a radius of a graph (minimum of longest paths) is not adequate.
> 4. Regarding lines 285-288, the average actual time it took to run the Python code on a single core, without any software or hardware-specific optimization, is as follows. Run-times for (FCI, ICD) are 15 nodes: (0.5, 0.4), 20 nodes: (8.6, 5.2), 25 nodes: (60, 21), and 35 nodes: (2041, 132) *seconds*. Please note that in the paper we calculated the ratio for each tested graph separately and reported the average. We will clarify this in the paper.
> 5. We recorded the CI tests performed by each algorithm, and counted the number of CI tests per conditioning set size. We did not restrict the conditioning set sizes or intervene during the execution of the algorithms.
>
> *Response to other comments*
>
> - We will formally describe the problem of anytime learning of causal graphs in the sense of learning *n*-O-equivalence classes.
> - We will add "every possible" in line 80.
> - We will write "sub-path" in stead of subpath. In line 95 we will state that the referenced sub-path is a sub-path of the considered path.
> - We will explicitly state which rules we used in line 17 of Algorithm 1.

---

> > ### Comment · Reviewer_WByU · 2021-08-30
> > **Response to authors**
> >
> > I would like to thank the authors for their detailed response. I would request them to include the response point (3) in their paper, as it helps clarify the nature of PDS-paths.

---

### Author Response · Authors · 2021-08-11
**To all reviewers**

We truly enjoyed reading the reviews, and appreciate the suggestions and detailed feedback, which help us to significantly improve the quality of the paper.

---

### Decision · Program_Chairs · 2021-09-27

**Decision:**

Accept (Poster)

**Comment:**

All reviewers are favorable after the author responses.

Clearly, this is a fairly significant contribution. Reducing the number of CI tests within the FCI family of algorithms (handling both selection bias AND confounders) is very important. I am impressed by the significant reduction in number of CI tests and significant improvement in orientation accuracy (which is non trivial) empirically. Further the anytime guarantees and associated theoretical results make it a solid paper. The authors even clarified key technical concerns after which reviewers felt more positive.

 Please do add experimental results quoted in the review response about FPR/FNR vs other FCI based algorithms and please do make changes as recommended by reviewers regarding rephrasing of Lemma 1.